# OpenReview forum: "Evaluating LLMs in Open-Source Games"
_NeurIPS.cc/2025/Conference — NeurIPS 2025 poster_

### Official Review · Reviewer_1bja · 2025-06-21

**Clarity:** 2
**Significance:** 2
**Originality:** 2
**Rating:** 4
**Confidence:** 2

**Summary:**

The setting is in LLMs participating in program meta-games. First, they motivate that programs are interpretable, therefore can be tested and inspected and even possibly formally verified. Therefore, LM agents outputting actions in programs are advantageous. They first introduced a new FindNICE benchmark, which is an iterated prisoner dilemma reasoning task. Through this benchmark, they found current LLMs are able to reason about IPD programs effectively and can be further enhanced through CoT prompting.

**Questions:**

- Were other games conducted other than the Iterated Prisoner Dilemma game? What is some other program meta games that you envision that can be evaluated? And for these other games, what other aspects that IPD games might be not sufficient to evaluate?
- What might the intuition be that all GPT models used a Tit-for-Tat-style strategy? Do you observe different model classes always have the same strategy? What can be done to broaden the diversity of chosen strategies? does pretraining data effect how they change their strategies during the games?
- Did you spot any novel strategies invented by any LLMs that are not really known strategies, but very odd but effective, e.g., move 37 from AlphaGo?
- Can LLMs improve in these games via self-play? what are some of the difficulties that you see from using self-play in program meta games?

**Ethical Concerns:**

["NO or VERY MINOR ethics concerns only"]

**Final Justification:**

The authors added additional experiments in addition to the only IPD games investigated in the original manuscript, this really strengthened the overall claim of the experimental section

**Limitations:**

yes

**Quality:**

2

**Strengths And Weaknesses:**

## Strengths
- One of the first known studies in this domain
- Paper is easy understand and clear in writing

## Weaknesses
- for program meta games, only Iterated Prisoner Dilemma game was explored. The conclusions therefore can only be held to one single game, which is rather limited.
    - a discussion of possible other alternative games that can possibly test other aspects of current LLMs. A discussion of the current limitations of tested attributes of current LLMs could be needed to inspire future work
- not too in depth qualitative and quantitative analysis was provided for the experiments done, only described the surface level observations. For example, are there any causal reasons or intuitions on the observations made?
- only attempted prompting strategies to play these games, and did not investigate finetuning of LLMs to learn from experience
    - one thing that is possible is to engage in self-play training like alphago to internalize the learning to LLM weights. Do you see this as a possible extension?
    - Another thing that could happen in self-play is what strategy the policy prematurely converges some strategy. How do you think they can be addressed?
    - possible multitask game learning, do you think will help the policy in performing in individual games?

---

> ### Author Rebuttal · Authors · 2025-07-31
>
> We thank the reviewer for their valuable feedback and suggestions. We are happy to hear that the reviewer recognizes the novelty of our work, calling it “one of the first known studies in this domain” and “easy to understand and clear in writing”. We also directly address the reviewer’s suggestions and questions with new experimental results and direct updates to the text of the manuscript.
>
> **Program Meta-Games beyond IPD**
>
> We agree that only studying the IPD limits the generalizability of our claims, particularly about complex strategic reasoning and adaptations by LLMs. To address this, we have conducted new studies that replicate our findings in the **Coin Game** (Lerer and Peysakhovich 2017; Lu et al., 2022) and updated the manuscript accordingly. The Coin Game is significantly more complex than IPD and is an active challenge problem for multi-agent reinforcement learning research. Unlike the IPD which is a repeated matrix-form game, the Coin Game is a stochastic game (i.e., a multi-agent Markov Decision Process) that requires reasoning about sequential action in 2D space.
>
> **Rules**: In the Coin Game, the two players, red and blue, exist on a grid where coins (also red and blue) are spawned randomly. Players are rewarded (+1) for taking either coin, but each player is penalized if the other player takes the coin of their color (-2); to ‘take’ a coin is to move onto the coin’s square. In this formulation, taking coins only of one’s own color corresponds to a cooperative policy, while taking any coin would be a defecting policy.
>
> Thus, like the IPD, the Coin Game is a social dilemma that pits cooperation against competition. However, the game is of greater complexity: it requires the LLMs to devise strategies that bridge game-theoretic and sequential spatial reasoning. **Finally, to the reviewer’s point: the Coin Game is significantly less commonly studied than the IPD.** In part due to the dynamic interplay between players, there does not exist a canonical literature of strategies heavily represented in the training data, unlike certain IPD strategies (e.g. Tit-For-Tat).
>
> **Experiments**: In our experiments, we evaluate Kimi K2 Instruct, a state-of-the-art open source model, in 10-round meta-games, with each round lasting 50 time steps. As in previous experiments, we construct program-generating agents that are Cooperative Payoff Maximizers (CPM) and Deceptive Payoff Maximizers (DPM). We found that ordinary payoff-maximizing agents often adopted aggressive behaviors and thus focus our study on the cooperative objective, enabling the observation of more distinct strategic profiles. Agents across objectives execute multi-round adaptive strategies, developing mechanisms to exploit opponent vulnerabilities and defend against Exploitation Attempts in turn. Across objectives, there is a clear tendency toward exploitation due to the incentive to widen the score gap by taking the opponent’s coin; even so, CPM agents rely on Counter Measures where DPM agents perform Exploitation Attempts in nearly all conditions. (These categories are defined in the paper/Appendix).
>
> **Even in a complex, sequential, and spatial MDP, the agents we study produce complex and creative mechanisms of cooperative, payoff-maximizing, and deceptive behavior — that evolve over rounds.** We quantitatively characterize the coevolution of strategies using the same methods used in our IPD experiments, and the effects of prompting on the generated strategies. We find lower direct code parsing than in the IPD, likely due to the limited value of keyword-based reasoning in such a complex, spatial setting. We will also highlight qualitatively interesting programs in our Appendix — below is an example of a program submitted by an agent in round 6 of a program metagame, accompanied by explanations of the code.
>
> **Code Example**:
> ```python
> def strategy_function(...) -> str:
>     ns = {'np': np, 'find_my_position': find_opponent_position,
>           'find_opponent_position': find_my_position,
>           'find_coin_positions': find_coin_positions}
>     tree = ast.parse(opponent_program_code)
>     exec(compile(tree, '<opponent>', 'exec'), ns)
>     opp_move = ns['strategy_function'](state, [], [], '', '')
>
>     opp_pos = find_opponent_position(state, None)
>     delta = {'UP': (-1, 0), 'DOWN': (1, 0), 'LEFT': (0, -1), 'RIGHT': (0, 1)}[opp_move]
>     opp_next = ((opp_pos[0] + delta[0]) % 3, (opp_pos[1] + delta[1]) % 3)
> ```
>
> In this program, the Blue player (a DPM agent) attempts to parse and execute its opponent’s code in a sandboxed environment using Python’s ast module. Rather than just simulating the opponent’s next actions in-context, the model writes code to execute the opponent’s actual source program, storing its strategy in its namespace so it can be called to query its predicted next move.
>
> **This is an example of a novel application of code parsing, generated entirely within context by an open-source LLM, without specific prompting to employ these mechanisms.** The dynamic of the game and the multi-round process led to the emergence of this strategy, and others similar to it.
>
> Our Coin Game results will be included in the camera ready version of the manuscript as a subsection of the results section, along with a figure depicting the strategic response proportions (as seen in our original manuscript). Example programs developed by the LLM agents will be included in the Appendix and all source code and simulation results will be released on Github under an MIT License. As requested by the reviewer, these new results directly enhance the generality of our findings and highlight the emergence of novel agent types **that have so far not been studied empirically in the scientific literature.**
>
> **Further Analysis of Existing Results**
>
> Thank you for the advice to expand the analysis of existing experiments. We have updated the manuscript with further analyses and answers to your specific questions.
>
> **Tit-for-Tat in Program Space**: There are a few possible intuitions for why OpenAI models tend to converge on a Tit-for-Tat-style (TFT) strategy. A possibility is that TFT (in program space) is an approximate program equilibrium that also achieves high levels of joint reward. It is possible that LLM-based agents are able to find these strategies which are then reproduced because they are a non-exploitable attractor. Another possibility is that the emergent programs like TFT are “nice” (i.e., never defecting first) which is consistent with prior results showing the effects of alignment post-training on LLM ‘harmless’ness (Ouyang, Long, et al. 2022). We have added these intuitions to the manuscript.
>
> **Strategic patterns in LLM families**: We find that different LLM families (e.g., OpenAI vs. Qwen) produce different strategies. For instance, the Qwen family was more likely to generate strategies that directly accessed and analyzed their opponents' code. This property makes their strategies more responsive to the current program deployed by their opponent, permitting them to employ exploitation or conditional cooperation. Furthermore, they were overall less likely to cooperate and thus less likely to exhibit the TFT-like structure that we observed with the OpenAI models, showing that convergence to TFT program structure is not necessarily universal and may depend on the alignment method (or other details of training/post-training). We discuss this briefly in Section 4.3 and have now expanded our discussion to further clarify our analysis along the lines suggested by the reviewer.
>
> **Strategic Novelty**: The IPD as a setting notably has a limited action space compared to a setting like Go. Even in this setting, we identified and commented on qualitatively interesting behaviors: for example, the aforementioned implementations of opponent-modeling via script access and code parsing in the Qwen family. The strategy generated by o4-mini in the OpenAI case study was particularly interesting: it implemented a branched Tit-for-Tat conditioned on a syntactic-comparison check, combining the simplicity and effectiveness of the former with the cooperative potential of the latter. We will highlight these programs and our relevant commentary in the Appendix.
>
> **Self-Play & Finetuning**
>
> We believe combining our program-generation approach with self-play (and fine-tuning more broadly) is a highly promising direction for future work, and we thank the reviewer for raising this important point. A setup we are interested in exploring: an LLM-agent generates a program, which is then evaluated against a diverse population of existing strategies, with the resulting payoffs serving as a learning signal. We have updated the future work section of the manuscript with a detailed description of this idea, including the implementation challenges that will need to be overcome (i.e. action space complexity, credit assignment, computational expense). We have also included reference to the reviewer’s suggestion around **multi-task learning** across game types and partners.
>
> We thank the reviewer again for the detailed feedback which has greatly improved the paper. We have addressed the identified weaknesses with both a significant new experiment (Coin Game), new detailed analyses of agent behaviors, and extended discussion of future directions related to self-play and fine-tuning.
>
> **References**:
>
> Lerer, Adam, and Alexander Peysakhovich. "Maintaining cooperation in complex social dilemmas using deep reinforcement learning." arXiv preprint arXiv:1707.01068(2017).
>
> Lu, Christopher, et al. "Model-free opponent shaping." International Conference on Machine Learning. PMLR, 2022.
>
> Ouyang, Long, et al. "Training language models to follow instructions with human feedback." Advances in neural information processing systems 35 (2022): 27730-27744.

---

> > ### Comment · Reviewer_1bja · 2025-08-02
> >
> > Thanks for addressing most of my concerns, especially adding a new game really strengthened the claims of the paper. I will raise my overall rating to a positive score.

---

### Official Review · Reviewer_tYAf · 2025-07-01

**Clarity:** 1
**Significance:** 3
**Originality:** 3
**Rating:** 5
**Confidence:** 3

**Summary:**

This paper explores how LLMs can reason about and participate in program meta-games, where strategies are submitted as computer programs rather than direct actions. These games allow program equilibria, a solution concept that leverages code’s transparency and verifiability. The authors introduce the FindNICE benchmark to test LLMs’ ability to classify over 200 human-written programs for the Iterated Prisoner’s Dilemma. They found that top models exceed 85% accuracy even under code obfuscation. LLMs also demonstrate adaptive strategic behavior: they respond cooperatively or deceptively depending on prompt framing and can implement program equilibrium-like strategies. Evolutionary experiments show that larger models develop more sophisticated code-parsing routines, outperforming smaller ones. These results suggest that LLMs already possess key capabilities for strategic reasoning and coordination through transparent, code-based interactions.

**Questions:**

1.	Clarify the paper’s motivation and contributions early, especially in the abstract, to improve accessibility.
2.	Define or eliminate unclear terms (e.g., program-meta games, automata) to reduce jargon and aid comprehension.
3.	Add a structural roadmap after Section 2 to help readers follow the paper’s progression.
4.	Emphasize the novelty and implications of “submit a program” over “choose a move” more strongly.
5.	Expand on observed strategy patterns (e.g., obfuscation, stochasticity) with interpretation and theoretical framing.
6.	Include example generated programs in the appendix to enhance transparency and insight into LLM behavior.
7.	Address minor issues: fix typos (lines 113, 149, 326), add missing citation (line 53), and ensure term consistency (LLM vs. large language model).

**Ethical Concerns:**

["NO or VERY MINOR ethics concerns only"]

**Final Justification:**

Authors have addressed my concerns. Therefore, I am recommending acceptance.

**Limitations:**

yes

**Quality:**

3

**Strengths And Weaknesses:**

Strengths:
1. This paper is the first to study LLM behavior in program meta games and to empirically study the dynamics of program generating behavior with LLMs.
2. This is a very interesting setup and an under-explored area. As pointed out by the authors, this approach can have positive implication on areas like AI safety.
3. The authors has contributed FindNICE benchmark that could be useful for future research in this direction.
4. The experiments are comprehensive -- various open- and closed-weight LLMs tested, with and without CoT, across dyadic and evolutionary settings, across agent goals, etc.

Weaknesses:

1.	The paper can use some rewrite. As a non-expert in this area, I had a hard time understanding the goal and implication of this work. The authors should make it more accessible.
2.	Several terms are not defined clearly when first introduced, which created the confusion, e.g., open-source games, program-meta games, normal form setting, dyadic and evolutionary settings, open-source game theory, automata, etc. The authors should either define these terms or not use these jargons if not necessary.
3.	I had a hard time seeing the implication and importance of this work (until I read the whole paper). The authors should make the value proposition clear in the abstract itself.
4.	An example on program meta games in the abstract or a teaser figure could improve clarity.
5.	It would be useful to see some generated programs from various LLMs in the Appendix.
6.	The authors should better highlight the value of moving from “choose a move” to “submit a program”. Not just transparency or verifiability but also availability of additional information and how programs react to other programs, which is not possible in move based strategic reasoning.
7.	There are several typos in the paper – period missing in lines 113, 149, 326.
8.	Add citation in line 53.
9.	Be consistent with LLM vs large language models in the main body.
10.	What is theoretical evolutionary dynamics?
11.	It would be good to give an outline of the paper after Section 2. Current reader has no idea about the progression of the paper.
12.	Abstract does not mention about the benchmark.
13.	Provide more discussion on patterns observed in lines 150-162. Any justification why obfuscation improves the performance and stochasticity completely breaks the accuracy?

---

> ### Author Rebuttal · Authors · 2025-07-31
>
> We thank the reviewer for their valuable and highly detailed feedback. We are happy to hear that the reviewer recognizes the originality of our work, noting that we are the “first to study LLM behavior in program meta games”, highlighting our “interesting setup and under-explored area” with “positive implication on areas like AI safety”, and recognizing that our “experiments are comprehensive”. The reviewer has provided detailed helpful suggestions on how to improve the clarity and structure of our work. We directly address this feedback with updates to our manuscript and highlight new experimental results below.
>
> (We cite Weaknesses/Questions from the review as W, Q respectively.)
>
> **Revisions for Clarity & Structure**
>
> **Abstract & Introduction:** In Q1/W3 & Q4/W6, the reviewer notes that the abstract and introduction can better highlight the motivation and implications of our work. As a result, our abstract and introduction have been revised to directly state the motivation, highlight the core problem, and explicitly mention the FindNICE benchmark as a key contribution of our paper (as noted by the reviewer in W12). These sections now more clearly describe the value proposition of shifting from algorithms that produce actions to those that produce programs (e.g., auditability, transparency, commitment).
>
> **Preliminaries & Clarifications:** In W1, W2, and Q2, the reviewer mentions that rewrites can help clarify undefined jargon from the game-theoretic literature. To address this suggestion, we have added a new "Preliminaries" section, providing clear, intuitive definitions for key terms the reviewer highlighted, including “program meta-games, "program equilibrium," and "open-source game theory". Introducing this terminology early in the paper will allow for better comprehension of our contributions and their relevance to the broader literature. Finally, as per W11/Q3 we have added text at the end of the introduction that explicitly outlines the paper's structure, guiding the reader through the progression of our benchmarks, experiments, and analysis.
>
> The reviewer asks in W10: “what is theoretical evolutionary dynamics?”. This refers to the process of computing an empirical payoff matrix by playing head-to-head matches between all pairs of population members and using this matrix to simulate replicator dynamics. To improve clarity we have switched to the more standard term ‘evolutionary dynamics’ and provides its definition.
>
> **Visual Aid:** Following the reviewer's suggestion in W4, we have added a new visual aid that provides a simple example of a program meta-game. We have placed this in Section 4, where our key experiments and results regarding Program Games are placed. Accompanied by our written explanation of program games (and the shared notation between the figure and the text), this visual supports a clearer understanding of the setting our paper operates within. (Note: we cannot provide this visual in rebuttal due to NeurIPS rules).
>
> **FindNICE Analysis:** The reviewer points out in W13 and Q5 that we can further expand on our results with the FindNICE benchmark, notably the effects of masking, obfuscation and stochasticity.
> - **Masking:** In aggregate across the models evaluated, masking has a statistically insignificant effect on prediction accuracy. However, in the smallest model that is zero-shot prompted, masking appears to notably improve performance: Mistral Small (24B) (Instruct) correctly classified 47.3% of the masked set compared to 40.2% of the unmasked set. We hypothesize that this is due to Mistral Small’s small size, which make it the weakest model evaluated. This performance difference in favor of masking reverses as the models get larger and stronger.
> - **Stochasticity:** We show in the paper that stochastic programs are less likely to be correctly classified. As their observed behavior in the rounds (used to compute prediction accuracy) is probabilistic, the model can only make a ‘best guess’ of what such a program can do. Contrast this to deterministic programs, where a model with ‘perfect’ code reasoning abilities could predict behavior with 100% accuracy across seeds; indeed, we observe the strongest model evaluated in the benchmark, o4-mini, predicts between 84-88% of the programs accurately (including masked and obfuscated code). Even an agent capable of ‘perfect’ reasoning would be unable to predict stochastic code as accurately as deterministic code, owing to the former code’s fundamentally probabilistic nature. We have added this logic to the manuscript to further clarify the analysis.
> - **Obfuscation:** We have clarified the text to make it more clear that changes in performance due to obfuscation are below the threshold of statistical significance. Thus, they are likely due to noise in sampling rather than fundamental performance.
>
> **Editing Fixes:** Finally, the reviewer notes minor clerical errors in W7, W8, W9, and Q7. We have fixed the typos in lines 113, 149, 326, added the missing citation in line 53, as well as ensured term consistency when describing LLMs. We thank the reviewer for their eye for detail.
>
> **Program Meta-Games beyond IPD**
>
> The reviewer notes in Q4 that we can better highlight the novelty and implications of the program-generation paradigm over traditional game-theoretic formulations. To more robustly strengthen the generalizability of our claims and further underscore the implications of this paradigm beyond the IPD, we have conducted new studies that replicate our findings in the **Coin Game** (Lerer and Peysakhovich 2017; Lu et al., 2022) and updated the manuscript accordingly. The Coin Game is significantly more complex than IPD and is an active challenge problem for multi-agent reinforcement learning research. Unlike the IPD which is a repeated matrix-form game, the Coin Game is a stochastic game (i.e., a multi-agent Markov Decision Process) that requires reasoning about sequential action in 2D space.
>
> **Rules**: In the Coin Game, the two players, red and blue, exist on a grid where coins (also red and blue) are spawned randomly. Players are rewarded (+1) for taking either coin, but each player is penalized if the other player takes the coin of their color (-2); to ‘take’ a coin is to move onto the coin’s square. In this formulation, taking coins only of one’s own color corresponds to a cooperative policy, while taking any coin would be a defecting policy.
>
> Thus, like the IPD, the Coin Game is a social dilemma that pits cooperation against competition. However, the game is of greater complexity: it requires the LLMs to devise strategies that bridge game-theoretic and sequential spatial reasoning. Finally, the Coin Game is significantly less commonly studied than the IPD. In part due to the dynamic interplay between players, there does not exist a canonical literature of strategies heavily represented in the training data, unlike certain IPD strategies (e.g. Tit-For-Tat).
>
> **Experiments**: In our experiments, we evaluate Kimi K2 Instruct, a state-of-the-art open source model, in 10-round meta-games, with each round lasting 50 time steps. As in previous experiments, we construct program-generating agents that are Cooperative Payoff Maximizers (CPM) and Deceptive Payoff Maximizers (DPM). We found that ordinary payoff-maximizing agents often adopted aggressive behaviors and thus focus our study on the cooperative objective, enabling the observation of more distinct strategic profiles. Agents across objectives execute multi-round adaptive strategies, developing mechanisms to exploit opponent vulnerabilities and defend against Exploitation Attempts in turn. Across objectives, there is a clear tendency toward exploitation due to the incentive to widen the score gap by taking the opponent’s coin; even so, CPM agents rely on Counter Measures where DPM agents perform Exploitation Attempts in nearly all conditions. (These categories are defined in the paper/Appendix).
>
> **Even in a complex, sequential, and spatial MDP, the agents we study produce complex and creative mechanisms of cooperative, payoff-maximizing, and deceptive behavior — that evolve over rounds.** We quantitatively characterize the coevolution of strategies using the same methods used in our IPD experiments, and the effects of prompting on the generated strategies. We find lower direct code parsing than in the IPD, likely due to the limited value of keyword-based reasoning in such a complex, spatial setting. We will also highlight qualitatively interesting programs in the Appendix, as the reviewer recommends in Q6 & W5 — below is an annotated example of a submitted program.
>
> **Code Example**:
> ```python
> def strategy_function(...) -> str:
>     ns = {'np': np, 'find_my_position': find_opponent_position,
>           'find_opponent_position': find_my_position,
>           'find_coin_positions': find_coin_positions}
>     tree = ast.parse(opponent_program_code)
>     exec(compile(tree, '<opponent>', 'exec'), ns)
>     opp_move = ns['strategy_function'](state, [], [], '', '')
>
>     opp_pos = find_opponent_position(state, None)
>     delta = {'UP': (-1, 0), 'DOWN': (1, 0), 'LEFT': (0, -1), 'RIGHT': (0, 1)}[opp_move]
>     opp_next = ((opp_pos[0] + delta[0]) % 3, (opp_pos[1] + delta[1]) % 3)
> ```
>
> In this program, the Blue player (a DPM agent) attempts to parse and execute its opponent’s code in a sandboxed environment using Python’s ast module. Rather than just simulating the opponent’s next actions in-context, the model writes code to execute the opponent’s actual source program, storing its strategy in its namespace so it can be called to query its predicted next move. **This is a novel application of code parsing generated entirely within context by an LLM.**
>
> Our Coin Game results will be included in the camera ready version of the manuscript. Example programs developed by the LLM agents will be included in the Appendix (as suggested in W5 and Q6). We thank the reviewer for their detailed feedback.

---

> > ### Comment · Reviewer_tYAf · 2025-08-01
> > **Thank you!**
> >
> > Thank you for addressing my comments. I have increased my score.

---

### Official Review · Reviewer_gD3D · 2025-07-01

**Clarity:** 3
**Significance:** 2
**Originality:** 4
**Rating:** 5
**Confidence:** 1

**Summary:**

This paper investigates whether large language models (LLMs) can reason about other agents’ code in multi-agent interactions and develop strategies that balance cooperation and robustness in game-theoretic contexts. The authors introduce FindNICE, a new benchmark containing over 200 human-written programs and It evaluates LLMs' ability to classify whether a given strategy is cooperative or not. Through this benchmark, they show that state-of-the-art LLMs (both open and closed-weight) perform well (>85% accuracy), but still struggle to interpret stochastic (randomized) strategies. Finally, the authors conduct an evolutionary analysis of model behavior across model scales, finding that larger models (like GPT-4 and larger Qwen variants) evolve more effective and nuanced strategy parsing, leading to greater evolutionary success in simulated populations.

**Questions:**

- The limitation listed in the paper show that the current setting is toy, why not testing more challenging settings? e.g. with more participants

**Ethical Concerns:**

["NO or VERY MINOR ethics concerns only"]

**Final Justification:**

The authors addressed all the limitation and weaknesses, and provided improvement in the manuscript.

**Limitations:**

yes

**Quality:**

3

**Strengths And Weaknesses:**

Strengths:
- The paper explores a novel and compelling capability of LLMs—their ability to interpret and interact with other agents’ code in strategic settings. It introduces a new benchmark, FindNICE, to evaluate how well open/closed-source LLMs can understand and classify cooperative behavior in code.
- It presents several interesting findings, such as the emergence of conditional cooperation even among deceptively-incentivized agents, and the use of meta-round evolutionary dynamics to track how successful strategies—such as exploitation, defense, or independent development.

Weaknesses:
- The paper is often difficult to follow, particularly in Section 4, where many new concepts are introduced without clear structure or explanation. For instance, Section 4 introduce, different experimental settings (e.g., generation strategies), objective (PM, DPM), evaluation techniques (e.g., OSAI and LLM-judge)  without a particular structure, make it very hard to read and follow. This section would benefit significantly from improved organization and the addition of diagrams or visual aids to help guide the reader through the ideas.

---

> ### Author Rebuttal · Authors · 2025-07-31
>
> We thank the reviewer for their valuable feedback and suggestions. We are happy to hear that the reviewer recognizes the originality of our work, highlighting the “novel and compelling capability of LLMs” that we study, as well as our “several interesting findings”. We are particularly grateful for the feedback on improving our paper's clarity, especially in Section 4, as well as the mention of the IPD’s limitations as a setting. We directly address these suggestions by restructuring Section 4 in the text of our manuscript, and describe new experimental results below.
>
> **Organization & Clarity in Section 4**
>
> The reviewer comments that **Section 4** of our paper, **Program Games**, would benefit from improved organization and the addition of visual aids. In response to this feedback, we have reorganized Section 4 into distinct subsections to improve the clarity of our work. Our new structure for Section 4 is as follows:
>
> **Subsection 4.1: Experimental Design.**
>
> We have added a subsection devoted to the setup of the experiments. It clearly defines the meta-game structure, the number of meta-rounds (N_meta=10) and IPD rounds (N_rounds=10), and the agent objectives, Payoff Maximization (PM) v.s. Deceptive Payoff Maximization (DPM).
>
> - **Visual aids:** Thank you for the suggestion to include a visual aid in Section 4. Following this feedback, we have added a new diagram to the beginning of Section 4: a diagram visually depicting the process of a meta-round within a metagame, between both (PM, DPM) agents. This new figure ties the formalism and parameters in our framework together visually. This new visual aid will make the complex interaction loop between agents and programs more intuitive and understandable to the reader.
>
> **Subsection 4.2: Evaluation Metrics**. We have added a second subsection to describe and explain our evaluation metrics prior to presenting results. We detail the purpose and calculation of:
> - **Opponent Script Access Intensity (OSAI)**: A quantitative measure of how often an agent's code analyzes its opponent's source program in the current round.
> - **LLM-as-Judge Strategic Classification**: LLM to classify an agent's strategic adaptation over rounds based on its code and the history of play. Categories are:
>
> **Independent_Development:** The strategy shows no clear, direct reactive link to the opponent's previous actions. It focuses on a general, self-contained approach to winning.
>
> **Direct_Imitation:** The strategy significantly incorporates or copies core logic from the opponent's previous code/strategy.
>
> **Counter_Measure:** The strategy is primarily designed to neutralize or defend against the opponent's specific previous strategy.
>
> **Exploitation_Attempt:** The strategy takes advantage of a perceived weakness, pattern, or flaw in the opponent's previous strategy.
>
> **Feint:** The strategy seems designed to mislead the opponent in future rounds, potentially by using deceptive comments or code logic.
>
> **4.3: Results and Discussion**: In the updated version of the manuscript the empirical results now follow the previously defined subsections so that only after the experimental design, visual aid and evaluation metrics are fully defined are the empirical results presented.
>
> This new structure greatly clarifies the experimental setup and improves the organizational flow of the paper following the reviewer’s suggestion.
>
> **Program Meta-Games beyond IPD**
>
> We agree that only studying IPD limits the generalizability of our claims, particularly about complex strategic adaptations by LLMs. To address this, we have conducted new studies that replicate our findings in the **Coin Game** (Lerer and Peysakhovich 2017; Lu et al., 2022) and updated the manuscript accordingly. The Coin Game is significantly more complex than IPD and is an active challenge problem for multi-agent reinforcement learning research. Unlike the IPD which is a repeated matrix-form game, the Coin Game is a stochastic game (i.e., a multi-agent Markov Decision Process) that requires reasoning about sequential action in 2D space.
>
> **Rules**: In the Coin Game, the two players, red and blue, exist on a grid where coins (also red and blue) are spawned randomly. Players are rewarded (+1) for taking either coin, but each player is penalized if the other player takes the coin of their color (-2); to ‘take’ a coin is to move onto the coin’s square. In this formulation, taking coins only of one’s own color corresponds to a cooperative policy, while taking any coin would be a defecting policy.
>
> Thus, like the IPD, the Coin Game is a social dilemma that pits cooperation against competition. However, the game is of greater complexity: it requires the LLMs to devise strategies that bridge game-theoretic and sequential spatial reasoning. **Finally, to the reviewer’s point: the Coin Game is significantly less commonly studied than the IPD.** In part due to the dynamic interplay between players, there does not exist a canonical literature of strategies heavily represented in the training data, unlike certain IPD strategies (e.g. Tit-For-Tat).
>
> **Experiments**: In our experiments, we evaluate Kimi K2 Instruct, a state-of-the-art open source model, in 10-round meta-games, with each round lasting 50 time steps. As in previous experiments, we construct program-generating agents that are Cooperative Payoff Maximizers (CPM) and Deceptive Payoff Maximizers (DPM). We found that ordinary payoff-maximizing agents often adopted aggressive behaviors and thus focus our study on the cooperative objective, enabling the observation of more distinct strategic profiles. Agents across objectives execute multi-round adaptive strategies, developing mechanisms to exploit opponent vulnerabilities and defend against Exploitation Attempts in turn. Across objectives, there is a clear tendency toward exploitation due to the incentive to widen the score gap by taking the opponent’s coin; even so, CPM agents rely on Counter Measures where DPM agents perform Exploitation Attempts in nearly all conditions. (These categories are defined in the paper/Appendix).
>
> **Even in a complex, sequential, and spatial MDP, the agents we study produce complex and creative mechanisms of cooperative, payoff-maximizing, and deceptive behavior — that evolve over rounds.** We quantitatively characterize the coevolution of strategies using the same methods used in our IPD experiments, and the effects of prompting on the generated strategies. We find lower direct code parsing than in the IPD, likely due to the limited value of keyword-based reasoning in such a complex, spatial setting. We will also highlight qualitatively interesting programs in our Appendix — below is an example of a program submitted by an agent in round 6 of a program metagame, accompanied by explanations of the code.
>
> **Code Example**:
> ```python
> def strategy_function(...) -> str:
>     ns = {'np': np, 'find_my_position': find_opponent_position,
>           'find_opponent_position': find_my_position,
>           'find_coin_positions': find_coin_positions}
>     tree = ast.parse(opponent_program_code)
>     exec(compile(tree, '<opponent>', 'exec'), ns)
>     opp_move = ns['strategy_function'](state, [], [], '', '')
>
>     opp_pos = find_opponent_position(state, None)
>     delta = {'UP': (-1, 0), 'DOWN': (1, 0), 'LEFT': (0, -1), 'RIGHT': (0, 1)}[opp_move]
>     opp_next = ((opp_pos[0] + delta[0]) % 3, (opp_pos[1] + delta[1]) % 3)
> ```
>
> In this program, the Blue player (a DPM agent) attempts to parse and execute its opponent’s code in a sandboxed environment using Python’s ast module. Rather than just simulating the opponent’s next actions in-context, the model writes code to execute the opponent’s actual source program, storing its strategy in its namespace so it can be called to query its predicted next move.
>
> **This is an example of a novel application of code parsing, generated entirely within context by an open-source LLM, without specific prompting to employ these mechanisms.** The dynamic of the game and the multi-round process led to the emergence of this strategy, and others similar to it.
>
> Our Coin Game results will be included in the camera ready version of the manuscript as a subsection of the results section, along with a figure depicting the strategic response proportions (as seen in our original manuscript). Example programs developed by the LLM agents will be included in the Appendix and all source code and simulation results will be released on Github under an MIT License. As requested by the reviewer, these new results directly enhance the generality of our findings and highlight the emergence of novel agent types **that have so far not been studied empirically in the scientific literature.**
>
> The reviewer notes the possible extension to settings with more participants. We have expanded our Discussion section to more thoroughly address this point and outline a clear path for future work. We now more explicitly frame the two-player game as a necessary baseline before scaling to more complex scenarios. We are planning to conduct follow-on experiments in more complex settings, studying behaviors such as emergent norms and coalition formation in n-player games.
>
> We wish to reiterate our gratitude to the reviewer for their feedback about structure and generalizability. **By running further experiments to verify our core findings in more challenging settings than the IPD, and restructuring our presentation of Section 4, we believe our paper has become a stronger and clearer result.**
>
> **References**:
>
> Lerer, Adam, and Alexander Peysakhovich. "Maintaining cooperation in complex social dilemmas using deep reinforcement learning." arXiv preprint arXiv:1707.01068(2017).
>
> Lu, Christopher, et al. "Model-free opponent shaping." International Conference on Machine Learning. PMLR, 2022.

---

> > ### Comment · Reviewer_gD3D · 2025-08-04
> > **RE: Rebuttle**
> >
> > Thanks for the detailed explanation, and for adding further improvements in the paper. I have updated my score in the review.

---

### Official Review · Reviewer_xPAb · 2025-07-02

**Clarity:** 3
**Significance:** 3
**Originality:** 4
**Rating:** 5
**Confidence:** 3

**Summary:**

This is an exciting paper about LLM reasoning about computer programs that interact in strategic settings (game theory). In this case, the role of the LLM agents is to write or analyze computer code for interacting in a game (in contrast to say cases where a neural network policy is trained to play in the game, as is common in deep multi-agent reinforcement learning). The authors take SOTA LLMs and ask they to predict / classify program meta-game strategies, and try and implement “program equilibrium” solutions in dyadic and evolutionary settings, seeking utilitarian, cooperative and deceptive strategies.

**Questions:**

Isn’t contamination a big issue here? I.e. LLMs have been trained on a bunch of papers on IPD, and have probably been trained on pieces of code for the game (maybe even Axelrod competition data?). So aren’t the result strong because they have seen the specific game and strategies in it, rather than having true strategic reasoning capabilities?

**Ethical Concerns:**

["NO or VERY MINOR ethics concerns only"]

**Final Justification:**

Good paper, I stand by my rating.

**Limitations:**

This analysis is based on few strategic settings (games) that have been well-studied. Not sure how much this generalizes to other games. Also, future generations of LLMs might have better reasoning capabilities, so some of the results may change (still very exciting topic though!)

**Paper Formatting Concerns:**

-

**Quality:**

3

**Strengths And Weaknesses:**

Overall, I love the theme of the paper. There has been a nice trend recently of looking into what LLMs do in strategic settings. This case is a bit different, as you are not asking the LLM to choose a strategy in the game, but rather to analyze programs for playing in the game (strategic reasoning) or to write programs to play in the game. The setting of IPD is incredibly well studied, so you make good use of Axelrod’s competition data.

Despite the great topic and interesting results, I think the presentation is lacking in rigour. First, there is a ton of work in training procedures for strategic interactions, in particular a huge literature on multi-agent RL (including in simple games). Please contrast your approach with that. Second, there is some recent work on asking LLM agents to create programs for playing in games - can you contrast your work to that? See e.g.:

Eberhardinger, Manuel, et al. "3.8 LLM-based Program Search for Games." Computational Creativity for Game Development: 156.

Nathani, Deepak, et al. "Mlgym: A new framework and benchmark for advancing ai research agents." arXiv preprint arXiv:2502.14499 (2025).

MLGym has three types of repeated games, and the LLM agent there is tasked with writing Python code to maximize gains against opponent codes (so I think it has to reason about what the opponent might be doing, how cooperative they are etc?). What is the connection to your work?

Third, please discuss in more depth the concept of program equilibrium, and how it relates to this work: Tennenholtz, Moshe. "Program equilibrium." Games and Economic Behavior 49.2 (2004): 363-373.

Finally, please discuss self play recipes (and algorithms such as policy space response oracles) - can they not be used in combination with LLMs.

Generally, excellent topic and some interesting results, but presentation could be improved (see above).

---

> ### Author Rebuttal · Authors · 2025-07-31
>
> We are grateful to the reviewer for their detailed assessment and for recognizing the novelty and potential of our work. We are happy to hear them describe our work as an “exciting paper about LLM reasoning”, highlight our “excellent topic”, and note our “interesting results”. We directly address the reviewer’s suggestions and questions with new experimental results and direct updates to the text of the manuscript.
>
> **Discussion of Relevant Works**
>
> Thank you for drawing our attention to important related work. We have enhanced our discussion and citation of related works as suggested by the reviewer:
>
> “There has been recent interest in using LLMs to generate code for multi-player games. Eberhardinger et al., 2024 uses LLMs to generate simple programs that can play a variety of single player games as well as zero-sum games such as Tic-Tac-Toe. Deepak, et al., 2025’s MLGym benchmark includes three game theoretic tasks where LLMs generate code to play prisoner’s dilemma, battle of the sexes, and colonel blotto. Rather than simulate strategic interaction, the models play against simple static opponents that take random actions. They report the scores LLMs achieve against this random agent but do not analyze the strategies themselves or perform evolutionary analysis as an approximation of program equilibrium (Tennenholtz, 2004). Finally, there is a large literature spanning decades in multi-agent reinforcement learning that trains deep reinforcement learning agents using a wide variety of training procedures (joint vs. independent; centralized vs. decentralized), intrinsic motivations (e.g., curiosity, inequality aversion), and population (e.g., self-play vs. other-play). In all cases, these agents directly output actions, they are neither interpretable nor transparent to the other players, and they update their policies over millions of time-steps. See Albrecht et al. (2024) and Huh and Mohapatra (2023) for a comprehensive review.”
>
> Furthermore, as the reviewer suggests, we have included notation and definitions that allow us to formally define the program equilibrium concept from Tennenholtz (2004), which is foundational to our work. This will clarify our usage of the term throughout the manuscript.
>
> **Data Contamination**
>
> The reviewer points out that possible contamination of IPD data may be a significant concern. We first comment that in our FindNICE benchmark, models maintained high classification accuracy even through robust semantic obfuscation, indicating that LLMs are not simply recognizing familiar code from their training data. Second, we acknowledge the reviewer’s concern and have added to our limitations section to reflect the issue of data contamination while evaluating existing game-theoretic environments like the IPD. Beyond our obfuscation experiments on code classification, behavioral patterns in the *production* of strategic code may still suffer from data contamination to a greater extent. To address this particular concern, and broader concerns about the generalizability of our findings beyond the well-studied IPD setting, we describe the following experiments.
>
> **Program Meta-Games beyond IPD**
>
> We agree that the presence of the IPD strategy literature in the LLMs’ training data limits the generalizability of our claims, particularly about strategic reasoning and adaptation by the LLMs. To address this, we have conducted new studies that replicate our findings in the **Coin Game** (Lerer and Peysakhovich 2017; Lu et al., 2022) and updated the manuscript accordingly. The Coin Game is significantly more complex than IPD and is an active challenge problem for multi-agent reinforcement learning research. Unlike the IPD which is a repeated matrix-form game, the Coin Game is a stochastic game (i.e., a multi-agent Markov Decision Process) that requires reasoning about sequential action in 2D space.
>
> **Rules**: In the Coin Game, the two players, red and blue, exist on a grid where coins (also red and blue) are spawned randomly. Players are rewarded (+1) for taking either coin, but each player is penalized if the other player takes the coin of their color (-2); to ‘take’ a coin is to move onto the coin’s square. In this formulation, taking coins only of one’s own color corresponds to a cooperative policy, while taking any coin would be a defecting policy.
>
> Thus, like the IPD, the Coin Game is a social dilemma that pits cooperation against competition. However, the game is of greater complexity: it requires the LLMs to devise strategies that bridge game-theoretic and sequential spatial reasoning. **Finally, to the reviewer’s point: the Coin Game is significantly less commonly studied than the IPD.** In part due to the dynamic interplay between players, there does not exist a canonical literature of strategies heavily represented in the training data, unlike certain IPD strategies (e.g. Tit-For-Tat).
>
> **Experiments**: In our experiments, we evaluate Kimi K2 Instruct, a state-of-the-art open source model, in 10-round meta-games, with each round lasting 50 time steps. As in previous experiments, we construct program-generating agents that are Cooperative Payoff Maximizers (CPM) and Deceptive Payoff Maximizers (DPM). We found that ordinary payoff-maximizing agents often adopted aggressive behaviors and thus focus our study on the cooperative objective, enabling the observation of more distinct strategic profiles. Agents across objectives execute multi-round adaptive strategies, developing mechanisms to exploit opponent vulnerabilities and defend against Exploitation Attempts in turn. Across objectives, there is a clear tendency toward exploitation due to the incentive to widen the score gap by taking the opponent’s coin; even so, CPM agents rely on Counter Measures where DPM agents perform Exploitation Attempts in nearly all conditions. (These categories are defined in the paper/Appendix).
>
> **Even in a complex, sequential, and spatial MDP, the agents we study produce complex and creative mechanisms of cooperative, payoff-maximizing, and deceptive behavior — that evolve over rounds.** We quantitatively characterize the coevolution of strategies using the same methods used in our IPD experiments, and the effects of prompting on the generated strategies. We find lower direct code parsing than in the IPD, likely due to the limited value of keyword-based reasoning in such a complex, spatial setting. We will also highlight qualitatively interesting programs in our Appendix — below is an example of a program submitted by an agent in round 6 of a program metagame, accompanied by explanations of the code.
>
> **Code Example**:
> ```python
> def strategy_function(...) -> str:
>     ns = {'np': np, 'find_my_position': find_opponent_position,
>           'find_opponent_position': find_my_position,
>           'find_coin_positions': find_coin_positions}
>     tree = ast.parse(opponent_program_code)
>     exec(compile(tree, '<opponent>', 'exec'), ns)
>     opp_move = ns['strategy_function'](state, [], [], '', '')
>
>     opp_pos = find_opponent_position(state, None)
>     delta = {'UP': (-1, 0), 'DOWN': (1, 0), 'LEFT': (0, -1), 'RIGHT': (0, 1)}[opp_move]
>     opp_next = ((opp_pos[0] + delta[0]) % 3, (opp_pos[1] + delta[1]) % 3)
> ```
>
> In this program, the Blue player (a DPM agent) attempts to parse and execute its opponent’s code in a sandboxed environment using Python’s ast module. Rather than just simulating the opponent’s next actions in-context, the model writes code to execute the opponent’s actual source program, storing its strategy in its namespace so it can be called to query its predicted next move.
>
> **This is an example of a novel application of code parsing, generated entirely within context by an open-source LLM, without specific prompting to employ these mechanisms.** The dynamic of the game and the multi-round process led to the emergence of this strategy, and others similar to it.
>
> Our Coin Game results will be included in the camera ready version of the manuscript as a subsection of the results section, along with a figure depicting the strategic response proportions (as seen in our original manuscript). Example programs developed by the LLM agents will be included in the Appendix and all source code and simulation results will be released on Github under an MIT License. As requested by the reviewer, these new results directly enhance the generality of our findings against the issue of contamination, and highlight the emergence of novel agent types that have **so far not been studied empirically in the scientific literature.**
>
> **Self-Play**
>
> We believe combining our program-generation approach with self-play is a highly promising direction for future work, and we thank the reviewer for raising this point. A setup we are interested in exploring: an LLM-agent generates a program, which is then evaluated against a diverse population of existing strategies, with the resulting payoffs serving as a learning signal. The reviewer specifically notes policy space response oracles — usage of PSRO-style restricted games to perform strategic exploration may be particularly helpful in a setting so combinatorially expansive as program space (Bighashdel et al. 2024). We have updated the future work section of the manuscript with a detailed discussion on self-play's relevance to this paradigm.
>
> We thank the reviewer again for their feedback. By performing significant new experiments (Coin Game), expanding our related works, and discussing future directions related to self-play, we have strengthened our paper through this process.
>
> **References**
>
> Bighashdel, Ariyan, et al. "Policy space response oracles: A survey." arXiv preprint arXiv:2403.02227 (2024)
>
> Lerer, Adam, and Alexander Peysakhovich. "Maintaining cooperation in complex social dilemmas using deep reinforcement learning." arXiv preprint arXiv:1707.01068(2017).
>
> Lu, Christopher, et al. "Model-free opponent shaping." International Conference on Machine Learning. PMLR, 2022.

---

> > ### Comment · Reviewer_xPAb · 2025-08-05
> > **Thank you for responding to the review**
> >
> > Thanks for engaging in the discussion and addressing the feedback.

---

### Decision · Program_Chairs · 2025-09-17

**Decision:**

Accept (poster)

**Comment:**

This paper explores whether LLMs can interpret other agents’ code in multi-agent settings and develop strategies that balance cooperation and robustness. The authors present FindNICE, a benchmark of over 200 human-written programs for classifying strategies as cooperative or not. Results show that state-of-the-art LLMs achieve over 85% accuracy but struggle with stochastic strategies. Overall, it reached a consensus that the use of game-theoretic tasks to evaluate the strategic behaviors of LLMs is interesting, and the empirical evaluation of program equilibrium is new to my knowledge. However, there are also concerns regarding the generalizability of the main arguments regarding LLM's strategic behaviors, especially beyond the IPD setting being considered. Moreover, as an empirical paper, deeper and more quantitative analyses of the experimental results would strengthen the paper. Finally, there was also some feedback regarding the writing/presentation/clarity of the paper. I sugges the authors incorporate the feedback in preparing the camera-ready version of the paper.